# Fabrication of a Three-Dimensional Microfluidic System from Poly(methyl methacrylate) (PMMA) Using an Intermiscibility Vacuum Bonding Technique

**DOI:** 10.3390/mi15040454

**Published:** 2024-03-28

**Authors:** Shu-Cheng Li, Chao-Ching Chiang, Yi-Sung Tsai, Chien-Jui Chen, Tien-Hsi Lee

**Affiliations:** Department of Mechanical Engineering, National Central University, Taoyuan City 32001, Taiwan; lsc790391464@gmail.com (S.-C.L.); shoo6667@hotmail.com (C.-C.C.); t1995tony@gmail.com (Y.-S.T.); qwe40223@gmail.com (C.-J.C.)

**Keywords:** microfluidic chip fabrication, PMMA bonding, intermiscibility bonding technique, solvent miscibility, digital photoelasticity analysis

## Abstract

In this study, the fabrication of microfluidic chips through the bonding of poly (methyl methacrylate) (PMMA) boards featuring designed patterns to create a three-dimensional sandwich structure with embedded microchannels was explored. A key focus was optimization of the interface quality of bonded PMMA pairs by adjusting the solvent, such as such as acetone, alcohol, and their mixture. Annealing was conducted below 50 °C to leverage the advantages of low-temperature bonding. Because of the differences in the chemical reactivity of PMMA toward acetone, alcohol, and their combinations, the resulting defect densities at the bonding interfaces differed significantly under low-temperature annealing conditions. To achieve the optimal sealing integrity, bonding pressures of 30 N, 40 N, and 50 N were evaluated. The interface was analyzed through microstructural examination via optical microscopy and stress measurements were determined using digital photoelasticity, while the bonding strength was assessed through tensile testing.

## 1. Introduction

The emergence of microfluidic technology signifies a transformative period in the fields of science and engineering, particularly accentuated by global health crises such as the COVID-19 pandemic, which underscore the urgency for rapid and accurate diagnostic methods. Traditional diagnostic processes are often cumbersome and time-consuming, ill-suited to address such crises. Microfluidic systems, with their design and functionality, offer a promising alternative by significantly reducing the time and costs associated with the examination of liquid or gas samples. This efficiency is crucial not only for addressing immediate healthcare challenges, but it also plays a critical role in ongoing public health maintenance, enabling rapid monitoring of physiological conditions and potential pathogenic threats. Materials such as glass and silicon have been pivotal in manufacturing microfluidic devices, but polymers have gradually become the mainstream material for such devices due to their low cost and ease of processing. Among the most widely used polymers in MEMS devices are polydimethylsiloxane (PDMS), polymethyl methacrylate (PMMA), polystyrene (PS), cyclic olefin copolymers (COC), etc. These materials find applications in preventing the COVID-19 virus [1,2], lab-on-a-chip [3], blood testing [4], cell separation [5], biomedical sensors [6], environmental analysis [7], etc.

Poly (methyl methacrylate) (PMMA) plays a critical role in this evolution. At the core of these technological advancements lies PMMA, a material selected for its excellent optical transparency, outstanding processability, and compatibility with microfabrication techniques. Moreover, its ability to form microchannels through methods such as milling or laser irradiation [8], coupled with its polymer compatibility [9,10,11,12], renders PMMA an ideal choice for large-scale production of microfluidic devices. PMMA is recognized for its versatility and efficiency, serving as the foundational material for the manufacture of microfluidic devices. Bonding represents the final step in the fabrication of microchannel materials and is crucial for the integrity of the entire microfluidic device. Therefore, research focused on reducing bonding costs, saving time, and enhancing sealing can significantly expand the application of polymer microfluidic devices.

Currently, PMMA bonding mainly consists of two methods: direct bonding and indirect bonding, with direct bonding widely used due to the advantage of high optical clarity at the bonding interface. Direct bonding includes thermal bonding [13,14], UV or microwave-assisted bonding [15,16,17,18], as well as solvent bonding [19]. In 2009, Mathur et al. [20] began using PMMA for thermal compression bonding to manufacture microfluidic channels and devices. However, thermal bonding has the drawback of causing channel deformation. To achieve absolute contamination-free disposable components, the overall structure of microchannels, including the base and cover plates, must be completely sealed. Vrahatis et al. [21] proposed a low-temperature plastic bonding method for microchannel devices, enabling the production of low-cost disposable microchannels. Additionally, for UV-assisted bonding, the process requires expensive, high-intensity UV-C lamps since PMMA is not sensitive to wavelengths above 240 nm. UV-assisted bonding is a technique commonly used in microchannel fabrication for its precision and control, despite higher equipment costs and specialized processing requirements [22].

Today, solvent bonding is widely used due to its cleanliness, excellent sealing properties, high bonding strength, low cost, simplicity, and speed. The key to solvent bonding lies in considering various parameters (such as bonding area, bonding strength, microchannel cracks, different solvent ratios, etc.) to achieve the optimal solvent combination for PMMA bonding performance. Currently, many different solvent bonding methods for PMMA have emerged. For example, Bamshad et al. [23] found that a 70% isopropanol solution produced optimal results in less than 15 min and further cooling, achieving a maximum bonding strength of approximately 28.5 MPa, thereby achieving rapid and low-cost PMMA bonding. Mohammad M. Faghih et al. [19] discovered that using a mixture of 20% dichloromethane and 80% isopropanol, specific time and temperature conditions yielded the highest bonding strength (4.2 MPa), and pre-treatment with corona discharge before applying the solvent could enhance bonding strength and optical transparency. Zhang et al. [24] used a mixture of ethanol and chloroform as the bonding solvent, which could bond PMMA at low temperature and atmospheric pressure, with a bonding strength of up to 267.5 N/cm^2^ and microchannel deformation less than 7.26%. However, despite the promising findings regarding solvent combinations for PMMA bonding, certain limitations in the experimental approach warrant attention. The study lacks a comprehensive exploration of the long-term durability and stability of the bonded PMMA structures under varying environmental conditions. Additionally, the precise mechanism underlying the enhancement of bonding strength with corona discharge pre-treatment remains unclear and warrants further investigation to elucidate its effectiveness and reproducibility.

In this study, two solvents were used to bond PMMA in three different combinations, aiming to investigate their influence on bonding strength and microchannel microstructure. This analysis seeks to determine the optimal solvent combination. Looking ahead, PMMA’s role in microfluidic device fabrication is poised for further advancement and evolution. Interdisciplinary research efforts across materials science, engineering, and biotechnology are expected to reveal new fabrication methodologies and applications, potentially transforming various industries. Emphasis on sustainability and cost-efficiency in microfluidic device development remains paramount.

## 2. Experiment

The solvents used were pure alcohol, pure acetone, and a mixture of alcohol and acetone. After several tests (alcohol/acetone = 2:1, 1:1, and 1:2), a solvent ratio of 1:1 was found to result in the best performance. To fuse two PMMA surfaces, the PMMA surfaces were lightly solvated by a mixture of solvents at a certain proportion before contact. This process is called transesterification, in which the ester group R″ is replaced by an alcohol group R′. According to the specific reaction mechanism, the R′-group of R′-OH in an organic solvent replaces the R″-group on PMMA to generate new esters and alcohols. The reaction formula is shown in Figure 1 [25]. In these reactions, acid or base catalysts can also be added for catalytic reactions. The organic solvent used in the reaction is limited to alcohols or ketones, so this reaction is also called the alcoholysis reaction.

As shown in Figure 2, in the case of direct bonding, R′-COOR and R″-OH hydrogen bonds are formed between the two bonding surfaces. After sufficient annealing to form the preliminary bonds, the organic solvent R′-OH can diffuse out of the bonding surface. The size of the bonding surfaces decreases, and the original substrate ester R″-COOH is ultimately formed by bonding to obtain a complete bonding interface.

In the initial phase of the experiment, the preparatory step involved placing the lower cover of the specialized bonding jig securely within the confines of the designated experimental tank. This was followed by a carefully controlled procedure where the chosen bonding solvent was gently poured into the tank, ensuring the complete immersion of the previously washed PMMA microfluidic component. This step is critical as it prepares the PMMA surfaces for the bonding process by facilitating solvent interaction with the material’s surface.

Subsequently, with meticulous attention to precision, the upper cover of the bonding jig was positioned over the setup. This assembly was then secured using hexagonal socket screws, a choice dictated by their reliability in applying uniform pressure. The screws were methodically tightened, a process conducted in stages to gradually build up to the required pressure deemed optimal for achieving a robust bond between the PMMA components.

Following this, an essential verification step involved the thorough agitation of the assembly. This was performed to ensure the complete expulsion of any air bubbles trapped within the bonding interface, a critical factor in preventing potential weaknesses in the bonded structure. Once satisfied with the bubble-free state, the assembly was locked in place with a clamp, signifying readiness for the next phase of the experiment.

The entire assembly, now prepared and secured, was then placed within a vacuum chamber. The conditions within the chamber were meticulously controlled, achieving a vacuum of 10^−2^ Torr. This environment was maintained to facilitate the initial bonding phase, a precursor to further solidification processes.

The final step in the experiment involved the transition of the entire set of PMMA specimens to a vacuum oven. Within this controlled environment, the specimens were subjected to a temperature of 50 °C for a period of three hours. This post-bonding thermal treatment is crucial for enhancing the strength and durability of the bond by allowing the solvent to evaporate completely and the PMMA components to fuse effectively, thus ensuring the integrity of the microfluidic device’s structure.

This expanded narrative provides a comprehensive overview of the experimental procedure, emphasizing the systematic approach and technical considerations essential for achieving successful bonding of PMMA components in microfluidic device fabrication.

## 3. Experimental Procedure

For this experiment, we utilized double-sided polished PMMA sheets with dimensions of 60 mm × 60 mm × 1 mm. The sheets were first processed using a laser cutter to create microchannels on their surfaces. The specimens were then cut into 50 mm squares with a thickness of 1 mm and polished using 1 µm alumina. The specimens were cleaned by ultrasonic treatment in deionized water, followed by drying with nitrogen gas.

The experimental setup included a lower clamp plate inside an experimental tank. Various bonding solvents (95% ethanol, 99.87% acetone, and a 1:1 mixture of 95% ethanol and 99.87% acetone) were introduced into the tank. The upper clamp of the adhesive fixture was then aligned, and two PMMA samples were secured using an Allen wrench to achieve uniform bonding pressure. This setup was agitated gently to eliminate air bubbles. Subsequently, the entire assembly was placed in a vacuum chamber at a pressure of 10^−2^ Torr, followed by curing in a vacuum oven at 50 °C for 3 h.

The surface morphology of the specimens was characterized using optical microscopy. The bonding strength was evaluated with a universal testing machine (UTM), Figure 3 shows the tensile test program model.

## 4. Results and Discussion

The viscosities of the solvents we used, ethanol at 1.074 mPa·s and acetone at 0.306 mPa·s, indicate a notable difference that could initially influence the initial bonding capability through their effect on capillary [26]. Nevertheless, it is important to consider that following heat treatment, which promotes solvent evaporation and covalent bond formation, the bonding energy’s relationship shifts towards being more closely associated with the density and strength of the bonding interactions. This change suggests that the influence of solution viscosity on the final overall bonding quality becomes less significant.

### 4.1. Microscopic Observation of the Bonding Interface

The optical microscopy (OM) images presented in Figure 4, Figure 5 and Figure 6 offer a detailed visual comparison of the bonded PMMA specimens, highlighting the distinct outcomes attributable to the use of different bonding solvents under identical pressure conditions. These images serve as a critical tool in assessing the microstructural integrity and the occurrence of defects within the bonded PMMA specimens, providing invaluable insights into the solvent-mediated bonding process’s efficacy.

The employment of acetone as a bonding solvent was observed to result in significant cracking across the PMMA specimens. This detrimental effect is primarily ascribed to acetone’s potent dissolution capability, which, while effective in facilitating bonding, concurrently exacerbates the material’s susceptibility to structural compromises. Such observations underscore the critical balance between solvent strength and material integrity in the bonding process.

Conversely, specimens bonded using alcohol as the solvent demonstrated the presence of bubble infiltration within the bonded interfaces. This phenomenon can be attributed to alcohol’s relatively weaker solvent strength, which, although less likely to cause dissolution-induced damage, is insufficient to prevent the formation of voids and gaps, leading to compromised bonding quality.

A mixed solvent approach, incorporating both acetone and alcohol, yielded an intermediate microchannel morphology characterized by a notable reduction in defects. This strategy appears to harness the strengths of both solvents, mitigating the extreme outcomes observed with their individual use. The optimal microstructure was achieved at a bonding pressure of 30 N, indicating a delicate equilibrium between solvent composition and applied pressure. However, it was noted that elevating the pressure beyond this point, specifically to 40 N and 50 N, initiated minor cracking within the microchannels. This suggests a threshold in the pressure–solvent interaction beyond which the structural integrity of the PMMA begins to diminish.

These findings illuminate the intricate dynamics between bonding solvent selection, applied pressure, and the resultant microchannel morphology in PMMA specimens. The implications of this study extend to the optimization of solvent-mediated bonding techniques, emphasizing the need for a balanced approach that carefully considers both the solvent’s chemical properties and the mechanical pressures applied during the bonding process.

### 4.2. Tensile Testing

Figure 7, Figure 8 and Figure 9 illustrate the average tensile strengths measured in three tests conducted at 30, 40 and 50 N in different solvents. These data reveal that the PMMA specimens bonded with the mixed solvent consistently demonstrated the highest tensile strength at all tested pressures. Additionally, a trend of increasing tensile strength with increasing pressure was noted for each solvent, although the degree of increase varied.

The lower maximum tensile strengths observed when alcohol and acetone were used as solvents were attributed to their distinct interactions with PMMA. Bonding with alcohol resulted in bubble infiltration at the interface, leading to incomplete bonding and, consequently, a reduction in material strength. Acetone, while facilitating complete bonding, caused PMMA dissolution, leading to the formation of microcracks that compromised the structural integrity of the material. Conversely, the mixed solvent effectively harnessed the benefits of both acetone and alcohol, achieving an optimal balance that led to superior tensile strength outcomes.

## 5. Conclusions

Our research effectively demonstrated the feasibility of creating microchannels in PMMA by employing a range of bonding solvents under diverse pressure conditions, resulting in optimal low-temperature bonding outcomes. The bonding quality was thoroughly evaluated through tensile testing and optical microscopy, providing essential insights into the strength and integrity of the bonds formed under various experimental conditions.

The successful bonding of PMMA/PMMA relies on several critical factors:
Solvent effectiveness: A mixture of alcohol and acetone (1:1 ratio) was found to be the most effective bonding solvent. This mixture consistently led to the highest tensile strength and optimal microchannel integrity across all tested pressures (30 N, 40 N, and 50 N), outperforming the use of alcohol or acetone alone. For all solvents, an increase in bonding pressure generally led to enhanced rupture strength. However, at higher pressures (40 N and 50 N), this also resulted in greater deformation and, in some cases, more severe damage to the microchannels.Solvent-specific observations: While alcohol as a solvent resulted in significant bubble infiltration, leading to weaker bonding strength, it had the least impact on microchannel integrity at lower pressures. Acetone allowed complete bonding but at the expense of causing dissolution and fine cracks in the PMMA, especially at higher pressures. The alcohol–acetone mixture provided a balance, minimizing defects such as bubbles and cracks and thus maintaining the integrity of the microchannels while supporting strong bonding.Chemical compatibility: The chemical reactivity and compatibility of the chosen solvents with PMMA play pivotal roles. The solvent must be able to slightly dissolve or swell the PMMA surface to induce effective bonding without overly damaging the material.

In summary, this research underscores the importance of solvent selection and pressure control in the fabrication of PMMA-based microfluidic devices. The use of a mixture of alcohol and acetone as a solvent is a promising approach for achieving strong, defect-minimized bonding in microchannel fabrication, paving the way for more efficient and reliable microfluidic device production.

## Figures and Tables

**Figure 1 micromachines-15-00454-f001:**
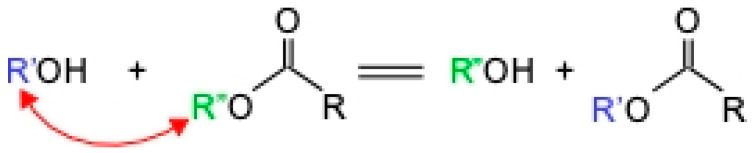
Transesterification chemical formula.

**Figure 2 micromachines-15-00454-f002:**
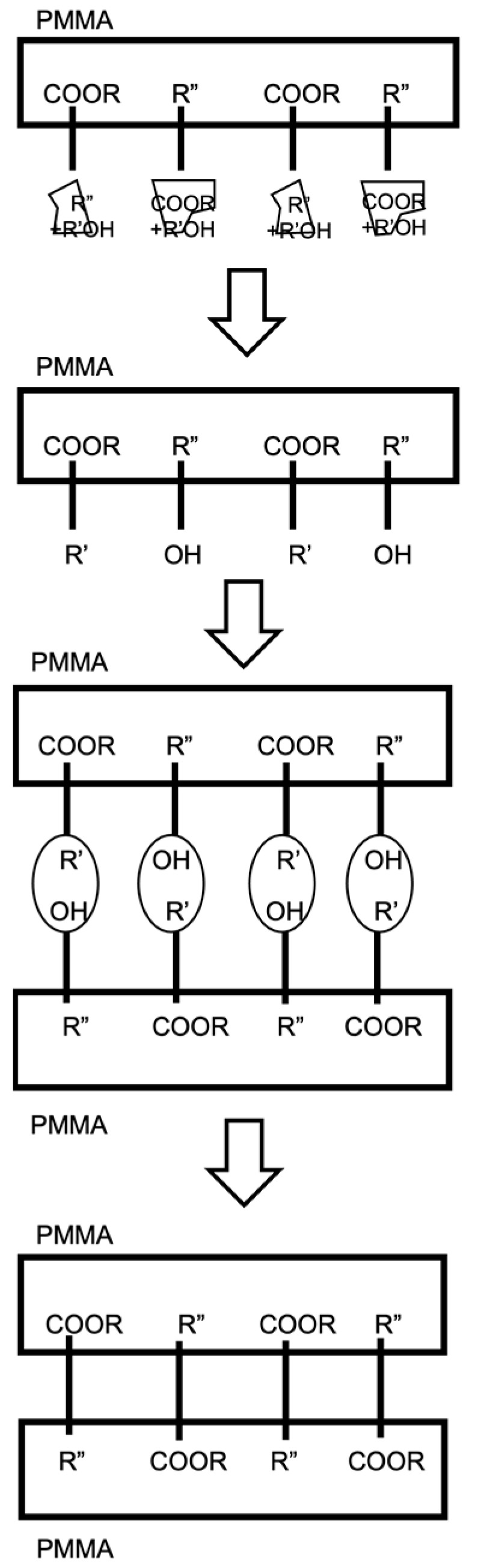
Schematic representation of chemical activation bonding in PMMA/PMMA. This diagram illustrates the step-by-step process of bonding PMMA to PMMA, highlighting the chemical radicals involved in the bonding mechanism, specifically COOR, OH^−^, and R^+^. Ref. [25] Prior to applying solvents for bonding, the bonding surfaces were mirror polished with 1 µm alumina powder to eliminate any scratches. Then, all PMMA specimens were cleaned with D.I. water.

**Figure 3 micromachines-15-00454-f003:**
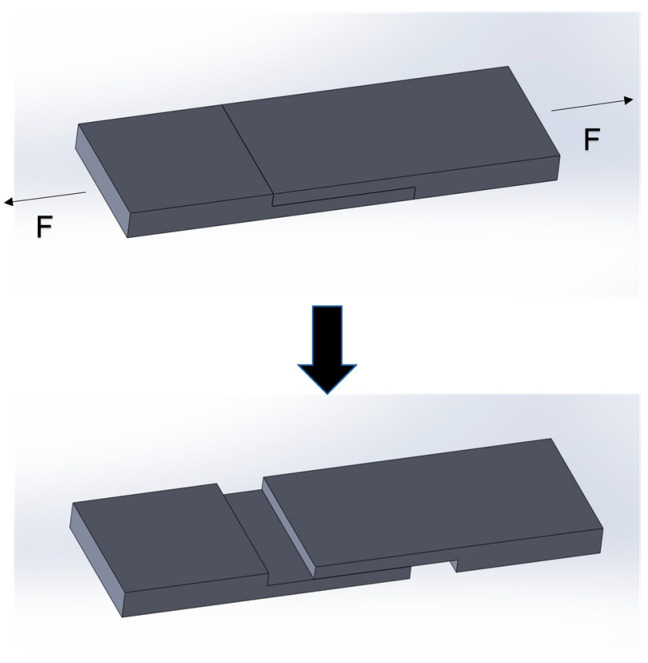
Tensile test program model.

**Figure 4 micromachines-15-00454-f004:**
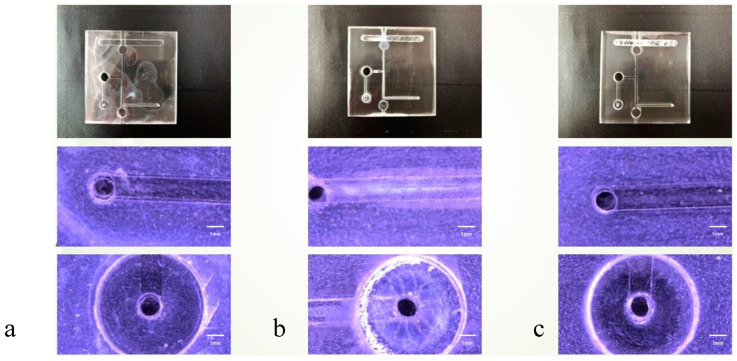
Illustration of microchannel conditions in PMMA Specimens under a bonding pressure of 30 N in various solvents. Optical microscopy (OM) 50× images demonstrating the effect of each bonding solvent: (**a**) alcohol, (**b**) acetone, and (**c**) an alcohol–acetone mixture. These images highlight the presence of bubbles at the PMMA bonding interfaces. Notably, compared with the individual solvents, the alcohol–acetone mixture resulted in a markedly reduced number of bubbles. (scalebar = 1 mm).

**Figure 5 micromachines-15-00454-f005:**
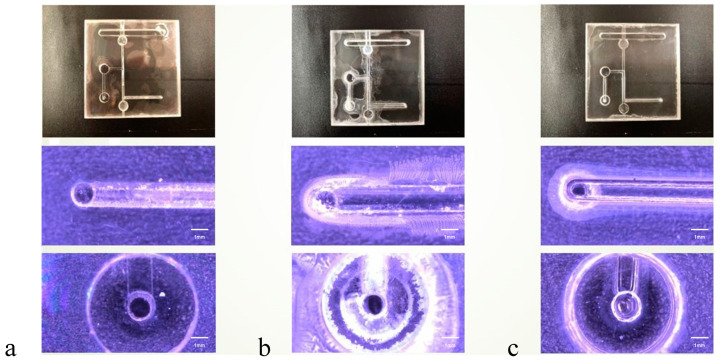
Microchannel status of PMMA specimens under a bonding pressure of 40 N in diverse solvents. Optical microscopy (OM) 50× images showing the impacts of each solvent: (**a**) alcohol, (**b**) acetone, and (**c**) a mixture of alcohol and acetone. Minor cracks are observable around the microchannels compared to the results obtained with a bonding pressure of 30 N (scalebar = 1 mm).

**Figure 6 micromachines-15-00454-f006:**
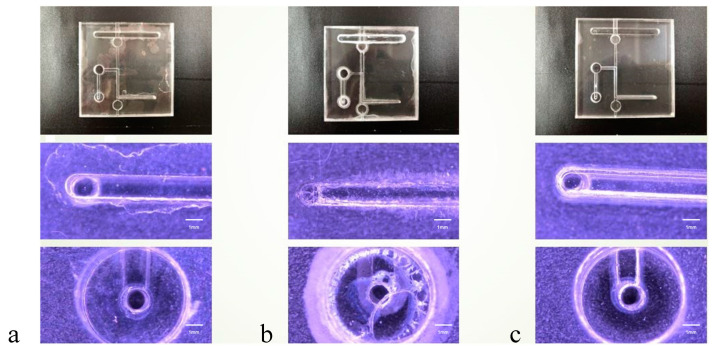
Microchannel conditions in PMMA specimens at a bonding pressure of 50 N in various solvents. Optical microscopy (OM) 50× images displaying the effects of (**a**) alcohol, (**b**) acetone, and (**c**) an alcohol–acetone mixture. Compared to the images obtained at a pressure of 30 N, these figures show an increase in the number of minor cracks and more severe damage around the microchannels (scalebar = 1 mm).

**Figure 7 micromachines-15-00454-f007:**
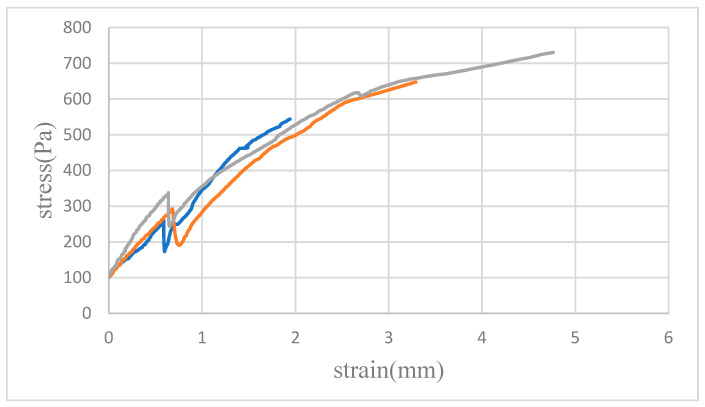
Tensile testing of bonded PMMA pairs and the results of measurements at a bonding pressure of 30 N. The graph shows three distinct curves corresponding to the solvents used: blue for alcohol, orange for acetone, and gray for the mixed solvent. Notably, the gray curve indicates enhanced ductility and superior rupture strength in comparison to those for the other solvents.

**Figure 8 micromachines-15-00454-f008:**
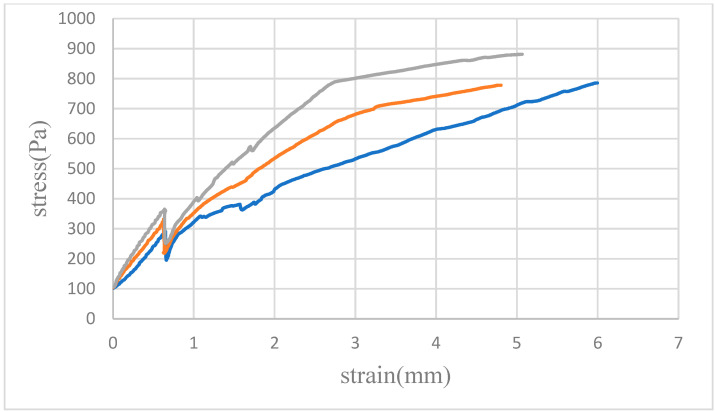
Tensile testing of bonded PMMA pairs and the results of measurements at a bonding pressure of 40 N. The graph displays three distinct curves for the solvents used: blue for alcohol, orange for acetone, and gray for the mixed solvent. Compared to the results obtained at a bonding pressure of 30 N, each curve here demonstrates not only an increase in rupture strength for all solvents but also a more pronounced deformation. Notably, the separation between the three curves is more substantial, indicating a clearer distinction in the performance of the sample in each solvent under increased bonding pressure.

**Figure 9 micromachines-15-00454-f009:**
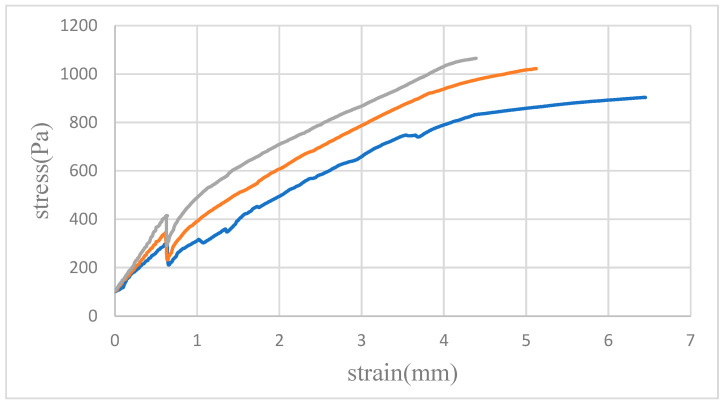
Tensile testing of bonded PMMA pairs and the results of measurements at a bonding pressure of 50 N. This graph features three distinct curves representing the solvents used: blue for alcohol, orange for acetone, and gray for the mixed solvent. Relative to the results obtained at a bonding pressure of 40 N, each curve indicates an increased rupture strength for all solvents. Notably, the gray curve exhibits the highest rupture strength, while the blue curve, corresponding to alcohol, shows the lowest rupture strength, potentially because of the higher bubble density.

## Data Availability

Data is contained within the article.

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
