# Peer review of "Fabrication of a Three-Dimensional Microfluidic System from Poly(methyl methacrylate) (PMMA) Using an Intermiscibility Vacuum Bonding Technique"

_micromachines, 2024, doi:10.3390/mi15040454_

Round 1
Reviewer 1 Report
Comments and Suggestions for Authors
The paper reported the fabrication of microfluidic chips through the bonding of PMMA boards with embedded microchannels. The interface was analyzed through microstructural examination via optical microscopy and stress measurements. There are some issues in the paper which need to be modified in order to improve the quality of the paper.
The comments are as follow:
1. Is there leakage problem, how to avoid it? The authors should conduct some experiments to verify it and add some discussions.
2. Figure 2 is too blurry to see. The scale bars are missing in Figure 3-5.
3. The article lacks a good introduction. The authors should add some background about various methods of chip fabrication, such as the bonding of PDMS and Glass, bonding of PMMA substrates…,
4 I recommend the authors to provide a more detailed introduction about the application of microfluidic chip. For example, cell manipulation, droplets, and POCT detection. 10.1039/d3lc00729d\ 10.1039/d2lc01193j\ 10.1002/celc.202300345\ 10.1002/admt.202301626\ 10.1039/d3lc00882g\ 10.1039/d3lc00851g \ 10.1021/acs.analchem.3c05755
5. Could this chip be reused again?
Author Response
- Is there leakage problem, how to avoid it? The authors should conduct some experiments to verify it and add some discussions.
Author's Response:
Thank you for your attention to the status of PMMA bonding. We would like to clarify that our experiments have consistently shown the integrity of PMMA bonding to be robust under a variety of conditions, including different pressures and applied solvents. The lowest recorded average tensile strength was approximately 543.8 kg/m2. This indicates a sufficient high bonding strength, leading us to confidently anticipate that leakage issues are unlikely under the tested conditions.
- Figure 2 is too blurry to see. The scale bars are missing in Figure 3-5.
Author's Response:
Thank you for the comment. We had improved Figure 2 and added appropriate scale bars in Figure 3-5. (Line 115 Page 3, Line 216 Page 6, Line 223 Page 7, and Line 228 Page 7)
- The article lacks a good introduction. The authors should add some background about various methods of chip fabrication, such as the bonding of PDMS and Glass, bonding of PMMA substrates…,
Author's Response:
Thank you for your insightful feedback. In response, we plan to enhance the manuscript by expanding the introduction to include a comprehensive overview of chip fabrication methods, including both PDMS and glass bonding, as well as PMMA substrate bonding. This addition will offer readers a solid foundation in the background and recent advancements in microfluidic chip fabrication. We aim to provide a concise yet thorough discussion on the principles, benefits, limitations, and applications of each method, ensuring a clear and informative context for understanding the significance of our research. (Lines 52-87 Page 2)
- I recommend the authors to provide a more detailed introduction about the application of microfluidic chip. For example, cell manipulation, droplets, and POCT detection. 10.1039/d3lc00729d\ 10.1039/d2lc01193j\ 10.1002/celc.202300345\ 10.1002/admt.202301626\ 10.1039/d3lc00882g\ 10.1039/d3lc00851g \ 10.1021/acs.analchem.3c05755
Author's Response:
Thank you for your valuable suggestion. We agree with the need to enrich the introduction with a more detailed exploration of microfluidic chip applications, such as cell manipulation, droplet generation, and point-of-care testing (POCT). Our goal is to offer readers a comprehensive view of the diverse and impactful applications of microfluidic technology. By doing so, we aim to underscore the relevance and breadth of our research within the broader context of microfluidics and its interdisciplinary uses. (Lines 36-40 Page 1)
- Could this chip be reused again?
Author's Response:
The reusability of the chip is influenced by its design, the materials used, and its intended applications. Our research focuses on refining the bonding process to maximize structural integrity. Given that our primary objective is to achieve strong and durable bonding, we consider PMMA chips to be ideally suited for single-use applications, thanks to their cost-effectiveness and minimal environmental impact.
Reviewer 2 Report
Comments and Suggestions for Authors
This manuscript describes a bonding technique for the fabrication of PMMA microfluidic devices. The authors investigated the effect of solvents and pressures on the device structures, The findings are interesting, but the amounts of data and discussions are limited (see comments). Thus, the quality of the manuscript seems not to reach the acceptance for this journal. This reviewer would recommend rejection.
Comments
1. There is no data on repeatability. The authors should repeat these experiments for proper discussions.
2. Although the authors discuss the cracking, there is no graph. The authors should prepare graphs with error bars for academic discussions.
3. The section on Materials and Methods is insufficient. For example, there is no information on how microfabrication of PMMA.
4. There was no description on why the design of microfluidic device was selected.
5. There is only few results. The authors should evaluate more the device (e.g. solution leaking).
6. Since there is no description of the purpose of the device design (CE? cell analysis?), this reviewer cannot understand whether the bonding is sufficient.
7. The scale bars should be added to Figs. 3-4.
8. There is no information on how to calculate “strain” in Fig. 7.
9. There is no unit of “strain” in Fig. 7. If it is dimensionless, “(-)” may be added.
10. “30 N, 40 N, and 50 N” might be changed to “30, 40, and 50 N”.
11. “Professor A. Mathur et al.” may be changed to “A. Mathur et al.”.
12. “density..” should be “density.”.
Author Response
- There is no data on repeatability. The authors should repeat these experiments for proper discussions.
Author's Response:
We appreciate your comment on the need for data on repeatability. In response, we have conducted the bonding experiments a minimum of three times to ensure the reliability of our results. Furthermore, the bonded microfluidic chips were successfully reproduced and utilized in blood investigations conducted by a collaborating research group. This repetition underscores the reproducibility and application readiness of our methodology, aligning with the standards required for collaborative research endeavors.
- Although the authors discuss the cracking, there is no graph. The authors should prepare graphs with error bars for academic discussions.
Author's Response:
Thank you for emphasizing the importance of graphical data in our analysis. Our initial examination centered on evaluating how different solvents influence the integrity of bonded PMMA microchannels, using optical microscopy for observations. While we understand the value of quantitatively detailing these findings, it's important to note that our discussion primarily targets the quality of bonding rather than crack formation per se. Additionally, our current laboratory setup lacks the specific equipment required to measure and graphically represent crack generation accurately. We are exploring ways to address this limitation and provide the most comprehensive data possible within our study's scope and resources.
- The section on Materials and Methods is insufficient. For example, there is no information on how microfabrication of PMMA.
Author's Response:
Thank you for highlighting the need for more detailed information in the Materials and Methods section, specifically regarding the microfabrication process of PMMA. To clarify, our manuscript details the process where microchannels are initially fabricated on PMMA sheets using a laser cutting technique. This is followed by a polishing step using 1 µm alumina particles, as described on line 160 of page 4. We appreciate your feedback and will ensure this process is clearly articulated and expanded upon in the revised manuscript to provide a comprehensive understanding of our methodology.
- There was no description on why the design of microfluidic device was selected.
Author's Response:
Thank you for your suggestion. We will address this issue by providing a more comprehensive introduction, emphasizing the advantages of PMMA material as a microfluidic device and highlighting the advantages of solvent bonding over other bonding methods. (41-51)
- There is only few results. The authors should evaluate more the device (e.g. solution leaking).
Author's Response:
Thank you for your suggestion. The purpose of this paper is to assess the quality of PMMA bonding (including strength and channel integrity) under different solvents and pressures. Concerning the issue of leakage, our findings indicate that PMMA bonding remains fully intact under varying pressures and solvents. Even with the lowest average tensile strength reaching approximately 543.802 kg/m2, leakage is not anticipated to occur.
- Since there is no description of the purpose of the device design (CE? cell analysis?), this reviewer cannot understand whether the bonding is sufficient.
Author's Response:
Thank you for your insightful feedback. We acknowledge the oversight regarding the lack of a clear description of the purpose of our device design, particularly regarding its intended application, such as CE (capillary electrophoresis) or cell analysis. We recognize the importance of providing context to assess the adequacy of bonding in our device. In light of your comment, we will enhance our manuscript by including a comprehensive description of the intended application of the device design. This addition will elucidate the relevance of our bonding approach to the specific application, aiding readers in understanding its significance. We value the thoroughness of your review and are committed to ensuring that our revisions effectively address this concern. Thank you for your constructive input.(Lines 37-40 Page 1)
- The scale bars should be added to Figs. 3-4.
Author's Response:
Thank you for your suggestion. It is crucial to accurately depict the size and dimensions of the features. We will promptly rectify this oversight by adding appropriate scale bars to the figures, facilitating precise measurements and interpretation of the presented microstructural features. (Line 216 Page 6, Line 223 Page 6, and Line 228 Page 7)
- There is no information on how to calculate “strain” in Fig. 7.
Author's Response:
Thank you for bringing attention to the matter of strain calculation related to Figure 7. We appreciate your understanding that the focus of our study on microfluidic devices does not prioritize strain as a critical measurement, and as such, we did not include detailed strain calculations in our findings. Acknowledging this, we believe that the specific calculations for strain are not essential for the objectives and conclusions of our research. We aim for our manuscript to clearly communicate relevant methodologies and results that directly contribute to the study's main goals and appreciate your guidance in ensuring our work is both concise and focused on its core contributions.
- There is no unit of “strain” in Fig. 7. If it is dimensionless, “(-)” may be added.
Author's Response:
Thank you for your suggestion. We recognize the importance of clearly presenting units in our figures to ensure the experimental results are easily understandable. The oversight of not including the unit for "strain" in Figure 7 will be addressed by indicating that it is dimensionless, denoted by "(-)". This adjustment will be made promptly to improve the clarity and comprehension of our findings, ensuring that all figures accurately convey the necessary information. (Line 245 Page 7, Line 250 Page 8, and Line 258 Page 8)
- “30 N, 40 N, and 50 N” might be changed to “30, 40, and 50 N”.
Author's Response:
Thanks for your suggestion, changes have been made to address this issue. (Line 233, Page 7)
- “Professor A. Mathur et al.” may be changed to “A. Mathur et al.”.
Author's Response:
Thanks for your suggestion, changes have been made to address this issue. (Line 55, Page 2)
- “density..” should be “density.”.
Author's Response:
Thanks for your suggestion, changes have been made to address this issue. (Line 264, Page 8)
Reviewer 3 Report
Comments and Suggestions for Authors
This work investigates the bonding of PMMA boards to create a three-dimensional sandwich structure with embedded microchannels for the development of microfluidic chips. Effects of bonding solvent and boning pressure on the microchannel states have been explored. So, I think this interesting and meaningful work can be publishable after major revision.
(1) Scale bar should be added in Fig.3, 4, and 5.
(2) A schematic diagram should be added to demonstrate the procedure of Tensile Testing.
(3) This work should discuss the effect on solution velocity on the bonded microchannel.
(4) Is it possible to fabricate a microchannel realize the simple functions of microfluidic chips? for example, microfluid mixing or droplet generation.
(5) Advantages of this method should be demonstrated in the manuscript.
Comments on the Quality of English LanguageMinor editing of English language required
Author Response
(1) Scale bar should be added in Fig.3, 4, and 5.
Response:
Thank you for highlighting the importance of including scale bars in Figures 3, 4, and 5. Accurate representation of size and dimensions is indeed essential for a clear understanding of the microstructural features presented. We acknowledge this oversight and will immediately address it by incorporating scalebars (= 1mm) into these figures. This amendment will enhance the accuracy of measurements and facilitate a more precise interpretation of the depicted features. Your feedback is invaluable to us in ensuring the completeness and clarity of our manuscript. (Fig.4、Fig.5、Fig.6)
(2) A schematic diagram should be added to demonstrate the procedure of Tensile Testing.
Response:
We value your suggestion regarding the importance of enhancing the reader's comprehension of the experimental procedures. Acknowledging this, we incorporate a schematic diagram that clearly outlines the tensile testing process. Your feedback plays a crucial role in improving the quality and clarity of our manuscript. (175、Fig.3)
(3) This work should discuss the effect on solution velocity on the bonded microchannel.
Response:
Thank you for suggesting the inclusion of a discussion on how solution velocity affects the bonded microchannel. The viscosities of the solvents used, ethanol at 1.074 mPa·s and acetone at 0.306 mPa·s, indicate a notable difference that could initially influence the initial bonding capability through their effect on adhesion. Nevertheless, it's important to consider that following heat treatment, which promotes solvent evaporation and covalent bond formation, the bonding energy's relationship shifts towards being more closely associated with the density and strength of the bonding interactions. This change suggests that the influence of solution viscosity on the overall bonding quality becomes less significant. We will expand upon this aspect in our manuscript to offer a more comprehensive understanding of the factors influencing bonded microchannel performance. (181-187)
(4) Is it possible to fabricate a microchannel realize the simple functions of microfluidic chips? for example, microfluid mixing or droplet generation.
Response:
Yes, the fabrication of microchannels does indeed enable the realization of basic microfluidic chip functionalities, such as microfluid mixing or droplet generation. For example, similar to the human body, where blood and tissue fluids carry cells of different sizes and quantities, microfluidic devices can be specifically designed for these requestions. By leveraging precise channel geometries and controlled flow conditions, it's possible to facilitate the separation and analysis of various blood components. This demonstrates the potential of microchannel-based designs to implement essential functions within microfluidic systems.
(5) Advantages of this method should be demonstrated in the manuscript.
Response:
Thank you for highlighting the importance of delineating the advantages of our method. In response, we've ensured that our manuscript comprehensively discusses how employing a solvent mixture of alcohol and acetone for bonding PMMA enhances both the tensile strength of the bonded interfaces and the integrity of microchannels. This is substantiated by the empirical data and observations presented in Figures 4 through 9. These not only demonstrate the effectiveness of the processing but also its contribution to advancing microfluidic device fabrication technology and highlight its practical advantages in the field.

Round 2
Reviewer 1 Report
Comments and Suggestions for Authors
The paper can be accepted.
Author Response
Thank you for your comments, which are helpful in improving the manuscript.
Reviewer 3 Report
Comments and Suggestions for Authors
I think this work is publishable in Micromachines.